# Compact vs. Linear: Effects of Forest Structure, Patch Shape and Landscape Configuration on Black Alder Macromoth Communities

Sara La Cava [1,2,*], Margherita Lombardo [3,*], Vincenzo Bernardini [1], Federica Fumo [4], Giuseppe Rijllo [1,2], Rosario Turco [1], Laura Bevacqua [1], Giada Zucco [1] and Stefano Scalercio [1,2]

1. Council for Agricultural Research and Economics (CREA), Research Centre for Forestry and Wood, 87036 Rende, Italy
2. National Biodiversity Future Center (NBFC), 90133 Palermo, Italy
3. Department of Computer, Modeling, Electronic, and System Engineering (DIMES), University of Calabria, 87036 Rende, Italy
4. Department of Biology, Ecology and Earth Sciences (DIBEST), University of Calabria, 87036 Rende, Italy
* Correspondence: sara.lacava@crea.gov.it (S.L.C.); margherita.lombardo@unical.it (M.L.)

**Abstract:** Landscape configuration and forest structure assume an increasing importance as determinants of animal communities. This paper focused on nocturnal Lepidoptera inhabiting alder patches in the Sila National Park, Italy. According to their shapes, patches were divided into linear and compact ones to disentangle the roles of forest structure and landscape configuration in determining the composition of nocturnal Lepidopteran communities at different observation scales. We used the Mann–Whitney test for medians and Shannon diversity, equitability, Fisher's alpha, and nestedness to test differences among moth communities. We found that compact patches inhabited richer and more abundant communities. The abundance-based Correspondence Analysis showed moth communities clustered according to woodlot shape, except a compact woodlot with a linear-like moth community because it was entirely surrounded by grasslands. Percentage of forested area and abundance and composition of communities were positively correlated at 50 and 200 m buffers, while correlations were absent at smaller and larger buffers. Our results demonstrated that a width of 50 m may not be sufficient to give proper functionality to the wooded area, at least for moths. As a consequence, planning of forest restorations should consider the importance of increasing the structural habitat continuity at larger scales.

**Keywords:** Lepidoptera; landscape ecology; riparian forest; *Alnus glutinosa*; natural park; beta-diversity; south Italy

## 1. Introduction

Forests have been managed by man from a long time ago to produce timber, and their surfaces have been reduced for the increasing demands of pastures and agricultural and anthropic lands [1,2]. Deforestation disrupts ecosystem equilibrium and threatens associated biodiversity [3–7], mainly because of habitat reduction and fragmentation, which modify patch shape and woodlot structure [8–13]. When a forest is highly fragmented, there is an inevitable increase in forest edges, leading to a higher margin of effect on species assemblages, and the structure and ecological processes of ecosystems near the ecotone are affected [14]. In addition, modifications of habitat quality and heterogeneity, as well as surrounding matrix attributes, are known to have significant effects on species occurrence and population size [15–19]. Proper forest management devoted to habitat restoration should ameliorate connectivity and reduce fragmentation through the establishment of an ecological corridor and increasing patch size, both with positive effects on biodiversity [20–23], with the latter favoring the presence of the more vulnerable core species [24].

Aside from landscape drivers, pure dendrometric parameters and spatial tree arrangement also affect the abundance and composition of animal and plant communities. Structural parameters of forests are relevant for animal and plant diversity [25], i.e., the age of trees affects lichen and bird diversity [26], tree species composition affects animal community composition [27–29], and tree density and basal area affects diversity in dry and tropical forests [30].

Among forest types, riparian ones and associated biodiversity are those that suffer more from severe deforestation because of the additive, detrimental effects of climate change [31], mainly due to the expected alteration of rainfall regimes. Fragmentation of riparian forests has been observed all around the world [32–34], and protected areas have not been spared. In the Mediterranean Basin, they are particularly vulnerable because climate change there is stronger than elsewhere, with temperatures and aridity increasing faster than the global average [35].

Several studies analyzed the effects of changes in patch structure and shape of wood-lots on biodiversity, such as birds and mammals [8,10], reptiles [36], amphibians [37], and invertebrates [9]. Among invertebrates, nocturnal Lepidoptera communities are often used to investigate the effects of changes in patch shape and sizes of forests [13,24], as well as bioindicators for agricultural intensification and forest quality [38–41]. It is known that abundance and richness of moth species are influenced by patch size, quality of woodlot, surrounding matrix attributes, and edge length [15–19]. As the size of a woodlot patch increases, the richness and the abundance of moth species increase for the addition of forest core species [24]. Moths occupy several environments, and many species are linked to forested habitats [42]. Those linked to riparian habitats have rarely been studied [23,43], especially in Italy, where knowledge increased only in the last decades [44–48]. Riparian habitats are essential to preserve many Lepidoptera species, some of which find this area suitable habitat against global warming-induced range shift [49]. In fact, despite the exclusive species not being among the most abundant ones, the sampled riparian forest is an important component of beta-diversity [47].

Black alder (*Alnus glutinosa* (L.) Gaertn.) forests of the Sila plateau, in the middle of Mediterranean Basin, are perfect models to study the effects of landscape structure on biodiversity hosted in riparian forests, as they are (i) threatened by severe climate change and (ii) mostly reduced to stripes surrounded by pastures and cultivated lands, with a significant edge length and potentially with a marked margin effect.

In this paper, we analyzed nocturnal Lepidoptera sampled by Leonetti et al. [47] in linear and compact alder woodlots in order to test the hypotheses that linear woodlots inhabit an impoverished version of moth communities inhabiting compact woodlots. The latter are supposed to host a well-preserved moth community also composed of core species, as the margin effect is supposed to be lower than in linear ones. Then, we assessed the role of landscape drivers and forest structure in shaping moth communities, providing suggestions to manage this vulnerable habitat.

## 2. Material and Methods

### 2.1. Study Area

The study area was entirely comprised within the Sila National Park, Calabria region, Italy (Figure 1). Most of the territory extended between 1100 and 1200 meters of altitude, reaching the highest altitudes with Mount Botte Donato (1928 m a.s.l.) and Mount Curcio (1768 m a.s.l.). The climate of the study area was warm temperate, with relatively humid summers typical of upland Mediterranean zones. The mean annual precipitation was around 1240 mm, with a mean monthly maximum of 180 mm recorded in November and a mean monthly minimum of 33 mm in July [50]. Snow cover usually occurred from December to April. The metamorphic and granite substratum of the plateau, since it has poor permeability, favored the surface flow of the abundant rainfall with the formation of a capillary network of watercourses, which flowed into four main rivers: Neto, Crati, Trionto, and Tacina. The landscape was mostly composed of woodlands with a predominance

of *Pinus nigra* Poir. Subsp. *calabrica* (Loud.) Cesca and Peruzzi forests at lower altitudes and *Fagus sylvatica* L. in the upper belt [51]. Woodlands were interrupted by grasslands and cultivated fields. Riparian woodlots only extended along water courses. In detail, sampled woodlots were selected along the Neto and Cecita riversides, with altitudes ranging between 1250 and 1397 m a.s.l.

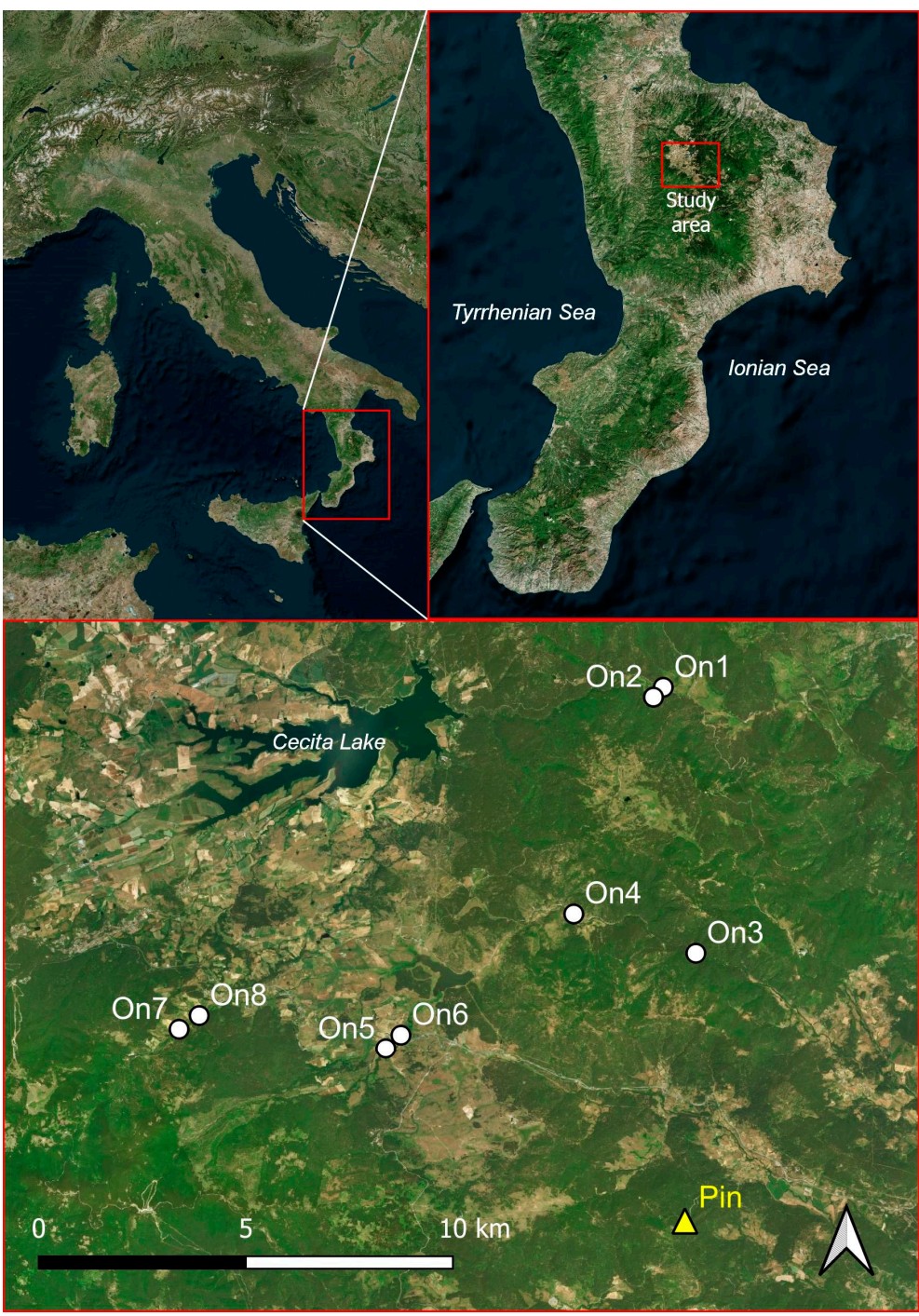

**Figure 1.** Localization of study sites. White circles indicate alder woodlots, and the yellow triangle indicates the pine woodlot.

Eight sites were monitored by Leonetti et al. [47], representative of riparian alder woodlots and of the main landscapes of the study area. Leonetti et al. [47] selected four pairs, each of them represented by a linear and a compact alder woodlot located neighboring

each other to minimize the bias due to local abiotic factors. Linear woodlots were those composed of alder trees growing along watercourses surrounded by open herbaceous habitats, whilst compact woodlots were defined as those at least 50 m wide with or without lateral continuity with other forest types (Table 1). One more site was monitored in this study within a pine woodlot for a better characterization of alder communities.

**Table 1.** Diversity and abundance of moths in sampled sites. Number of species (*S*); number of individuals (*N*); Shannon index (*H*); equitability (*J*); Fisher's alpha (*α*).

|  | *S* | *N* | *H* | *J* | *α* |
|---|---|---|---|---|---|
| Linear woodlots |  |  |  |  |  |
| On1 | 131 | 663 | 4.07 | 0.83 | 48.92 |
| On4 | 86 | 359 | 3.62 | 0.81 | 35.84 |
| On6 | 137 | 635 | 4.35 | 0.88 | 53.69 |
| On8 | 131 | 669 | 4.16 | 0.85 | 48.69 |
| Compact woodlots |  |  |  |  |  |
| On2 | 157 | 1179 | 4.17 | 0.82 | 48.63 |
| On3 | 151 | 1140 | 4.05 | 0.81 | 46.66 |
| On5 | 151 | 849 | 4.23 | 0.84 | 53.41 |
| On7 | 147 | 704 | 4.36 | 0.87 | 56.57 |

*2.2. Moth Sampling*

Moth data were mostly gathered from Leonetti et al. [47], supplemented by original data sampled in a pine woodlot and with taxonomic changes that occurred in recent years [52]. Due to the confirmed presence of both species in the study area [53] and their hard discrimination, we considered the recently recognized *Hoplodrina alsinides* and *H. octogenaria* as a species pair cited in this paper as *Hoplodrina* cfr. *octogenaria* (cfr. means compare).

Leonetti et al. activated a UV LED light trap per site one night per month from March to November 2017 in georeferenced points, obtaining a total of 72 samples. Traps worked simultaneously in all sites, reducing the effects of different weather conditions on collected data. Sampling nights were chosen near the new moon phase (±7 days), with temperatures no lower than the mean of the period, with low wind (<10 km/h) and possibly with no or low rain. Collected materials were sorted in the laboratory, and moths belonging to the selected superfamilies (Hepialoidea, Zygaenoidea, Cossoidea, Lasiocampoidea, Bombycoidea, Drepanoidea, Geometroidea, and Noctuoidea) were identified at the species level and counted. Voucher specimens were preserved in the scientific collection of Lepidoptera of the Research Centre for Forestry and Wood, Rende, Italy. For any further details on samplings, see Leonetti et al. [47]. We sampled a pine woodlot from April to October 2022 following the same trapping design, obtaining a total of 6 samples. Data from 78 samples were pooled and arranged in a species/site matrix and then submitted for statistical analyses.

*2.3. Forest Structure*

In this study, we selected structural parameters of the forest that are known to be relevant in determining animal and plant diversity [25], i.e., age of trees, which determines lichen and bird diversity [26] and tree density and basal area, which affect diversity in dry and tropical forests [30]. Furthermore, we separately evaluated these parameters for dead and living trees, as the former seems to affect forest diversity differently [54,55].

We evaluated classical dendrometric parameters within a radius of 25 m around sampling points in order to characterize alder woodlots. In detail, we measured separately for living and dead trees the following: estimated tree age, mean tree heights (m), tree density (n/ha), mean diameter of stems at breast height ($DBH_{mean}$), total basal area ($BA_{tot}$), and total dendrometric volume ($V_{tot}$). Tree heights were measured using an infrared ipsometer

Vertex III, whilst diameters were extrapolated from measured stem circumferences. Tree age for a plot was extrapolated by measuring the age of the tree with the DBH nearest to the mean of the plot by using a Pressler's coring device. Dendrometric volumes were computed by using a volume table for alder forests of the Sila Mountains [56].

### 2.4. Landscape Configuration

The landscape was described around each moth trap (sampling point) using circular buffers of 25, 50, 200, 500, and 1000 m. The smallest radius was the one describing the landscape from which the trap was expected to attract moths [57], and the largest was the one describing the landscape where most moths were expected to live, due to their dispersal abilities [58]. Resulting circular landscapes were analyzed using two landscape metrics: proportion of classes and shared edge length, both at a class level [59]. They were chosen among the others as the simplest measures of landscape composition and spatial configuration. Three classes were selected according to the study goals, including alder woodlots, forests, and grasslands (Figure 2). Alder woodlots were the subjects under study from which moths were sampled and analyzed. Forests, whichever their compositions were, shared similar abiotic conditions with alder woodlots, with higher permeability to moth movements than herbaceous habitats. Shared edge length was the measurement of the ecotone between forests and grasslands, becoming a proxy of habitat fragmentation in fixed buffers. Every single patch was manually digitized as a polygon in a GIS environment, based on Google satellite imagery (Map data ©2016 Google) uploaded through the QuickMapServices QGIS plugin and then merged by attributes. Feature areas and shared edges were automatically computed using QGIS (3.22 "Białowieża" release) processing tools.

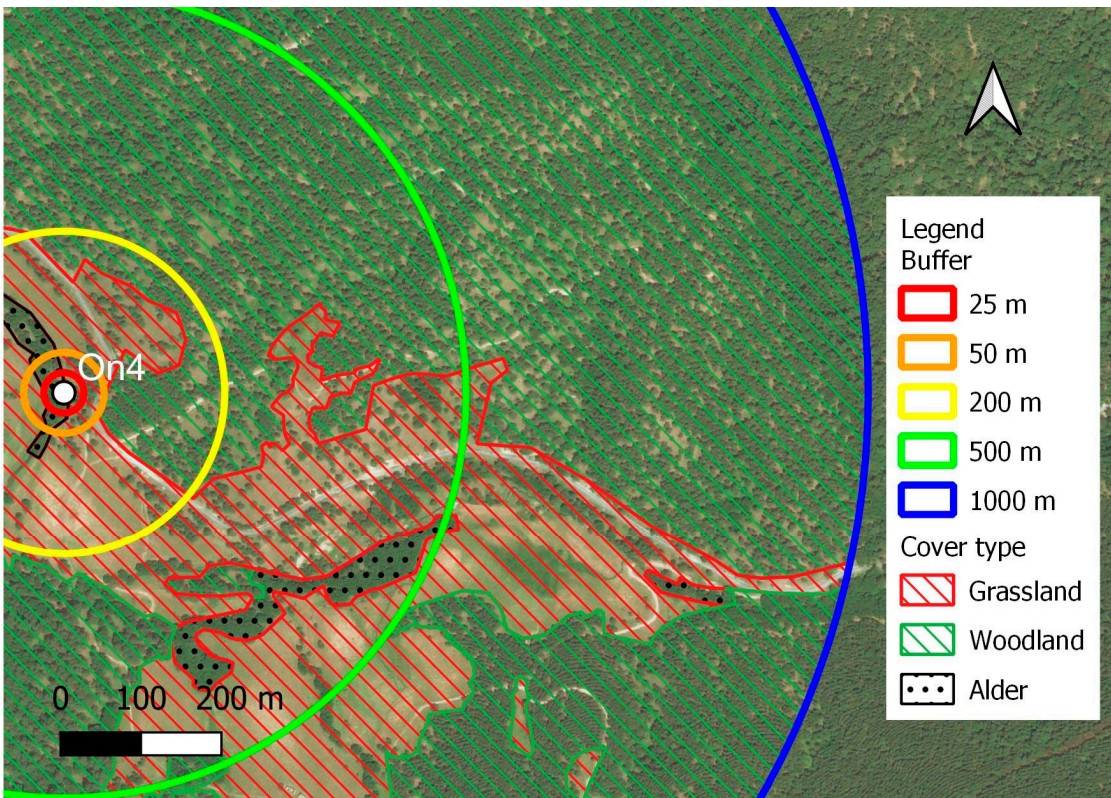

**Figure 2.** Schematization of landscape interpretation and analysis around light trapping points at five different buffers. The white dot indicates the location of the light trap.

### 2.5. Moth Analysis

Differences in abundance and richness of moth communities between linear and compact black alder woodlots were compared by means of the Mann–Whitney test for medians running PAST 4.03 [60]. Diversity indices were also computed, namely Shannon (*H*), equitability (*J*), and Fisher's alpha (*α*).

Main ecological processes shaping beta-diversity are known to be the turnover, or species substitution, and the nestedness, or species impoverishment. The diversity of altered habitats, as we hypothesized linear woodlots to be, usually tends to be nested within the natural ones, as compact woodlots were assumed to be in this study. As a result, we expected that in woodlot pairs, linear alder moth communities would be nested within compact ones. To test this hypothesis, we computed the "Nestedness metric based on Overlap and Decreasing Fill" (NODF) for a binary matrix following Almeida-Neto et al. [61], running Nestedness for Dummies (NeD) [62].

Then, the abundance matrix of sampled alder and pine woodlots were submitted to Correspondence Analysis (CA) by running PAST 4.03 [60].

## 3. Results

### 3.1. Moth

A total of 8451 individuals belonging to 332 taxa were included in this study, of which 6198 individuals belonging to 309 taxa were collected in alder woodlots (Table S1). The most abundant species were *Orthosia incerta*, *Eilema lurideola*, *E. complana*, and *Hoplodrina* cfr. *octogenaria*, representing, altogether, 18% of the whole sample.

Linear woodlots inhabited communities with significantly less species ($p = 0.028$) and less individuals ($p = 0.030$) than compact ones, but computed diversity indices were not statistically different (Table 1). However, linear communities were not nested within the respective compact pair, with one exception only represented by On4 (Table 2).

**Table 2.** Nestedness metric based on overlap and decreasing fill (NODF) for sampled woodlot pairs.

| Alder Woodlots Linear vs. Compact | NODF | Nested | |
|:---:|:---:|:---:|:---:|
| On1 vs. On2 | 49.344 | NO | $p > 0.05$ |
| On4 vs. On3 | 45.554 | YES | $p < 0.001$ |
| On6 vs. On5 | 50.064 | NO | $p > 0.05$ |
| On8 vs. On7 | 50.177 | NO | $p > 0.05$ |

The abundance-based Correspondence Analysis (CA) showed that moth communities were primarily shaped by forest types, with the pine strongly separated from alder communities along Axis 1. On the other hand, alder communities were only slightly separated along Axis 1, with most of the compact alders having lower values than linear ones (Figure 3). A fairly better separation of alder woodlots was observed along Axis 2, assuming most of compact ones were negative, and all linear ones had positive values. CA showed that the supposed compact woodlot On5 inhabited a linear community despite its forest structure at stand level (Figure 3).

By splitting individual moth samples according to CA in two groups, we found that among the ten most abundant species in linear communities, only two were also among the ten most abundant species in compact communities, namely *Orthosia incerta* and *Hoplodrina* cfr. *octogenaria* (Table 3). The most characteristic species for linear communities were *Agrotis cinerea*, *Luperina testacea*, and *L. dumerilii*, whilst those characterizing compact communities were *Eilema lurideola*, *E. complana*, and *Lithosia quadra* (Table 3).

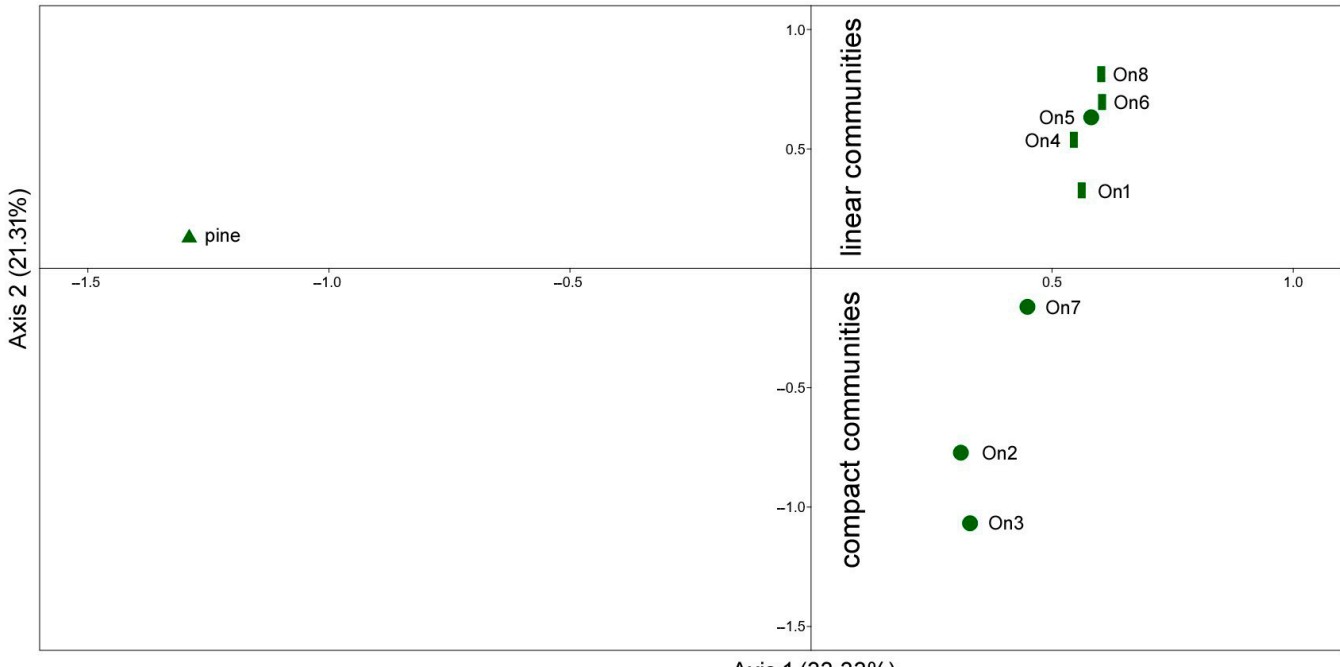

**Figure 3.** Moth communities sampled within sampled alder (circle: compact; bar: linear) and pine (triangle) woodlots plotted on a cartesian plan individuated by Axis 1 and Axis 2 of Correspondence Analysis.

**Table 3.** Abundance (N) and rank of the ten most abundant species within linear and compact black alder woodlots; cfr. means compare.

| Species | Linear Communities (*N* = 5) | | Compact Communities (*N* = 3) | |
|---|---|---|---|---|
| | *N* | Rank | *N* | Rank |
| Non-characteristic species | | | | |
| *Orthosia (Orthosia) incerta* | 229 | 1 | 103 | 4 |
| *Hoplodrina* cfr. *octogenaria* | 166 | 2 | 62 | 9 |
| *Peribatodes rhomboidaria* | 67 | 8 | 52 | 13 |
| *Epirrhoe alternata* | 65 | 9 | 23 | 34 |
| *Pachetra sagittigera* | 61 | 10 | 25 | 32 |
| *Lycia hirtaria* | 48 | 14 | 145 | 3 |
| Characteristic species for linear woodlots | | | | |
| *Agrotis cinerea* | 145 | 3 | 21 | 36 |
| *Luperina testacea* | 105 | 4 | 14 | 49 |
| *Luperina dumerilii* | 82 | 5 | 4 | 105 |
| *Mythimna (Mythimna) impura* | 79 | 6 | 16 | 44 |
| *Agrotis exclamationis* | 76 | 7 | 8 | 43 |
| Characteristic species for compact woodlots | | | | |
| *Eilema lurideola* | 19 | 44 | 289 | 1 |
| *Eilema complana* | 25 | 31 | 221 | 2 |
| *Lithosia quadra* | 5 | 106 | 96 | 5 |
| *Xestia (Megasema) triangulum* | 14 | 59 | 81 | 6 |
| *Diarsia mendica* | 20 | 43 | 64 | 7 |
| *Dysstroma truncata* | 2 | 155 | 64 | 8 |
| *Alcis repandata* | 10 | 73 | 60 | 10 |

### 3.2. Forest Structure

Among the parameters involved in the dendrometric analysis, only the tree density showed results driven by the woodlot choice. In fact, all compact woodlots were denser than linear ones in the analyzed buffer of 25 m, considering all tree species and alders alone (Table 4). Most of linear woodlots had older and larger trees than those composing compact ones, but they showed lower values of total basal area and total dendrometric volume (Table 4). The woodlot On1 was an exception among the linear ones as the one with the highest trees and the highest amount of total dendrometric volume. Similarly, On2 was an exception among the compact woodlots because of the very low total dendrometric volume. Dead trees were present in three compact woodlots and in one linear woodlot only. Tree species different from alders were present within two sites of both woodlot shapes but more abundant within compact ones. To summarize, linear woodlots were composed of less trees than compact woodlots, resulting in a lower amount of wood, but these trees were older and larger than those in compact woodlots.

**Table 4.** Forest structures of sampled woodlots. $DBH_{mean}$: mean $\pm$ S.D. of diameters at breast high; BA: basimetric area; $V_{tot}$: estimated total volume of wood.

| Shape of Alder Woodlots | Linear | | | | Compact | | | |
|---|---|---|---|---|---|---|---|---|
| **Woodlot** | **On1** | **On4** | **On6** | **On8** | **On2** | **On3** | **On5** | **On7** |
| All tree species | | | | | | | | |
| $Height_{mean}$ (m) | $35.9 \pm 3.9$ | $19.6 \pm 1.9$ | $18.9 \pm 3.2$ | $14.1 \pm 4.7$ | $20.6 \pm 1.7$ | $21.8 \pm 3.6$ | $16.2 \pm 3.7$ | $23.1 \pm 2.6$ |
| Estimated age (y) | 46 | 45 | 31 | 35 | 26 | 33 | 23 | 34 |
| Density (n/ha) | 589 | 287 | 597 | 342 | 835 | 916 | 1989 | 995 |
| $DBH_{mean}$ (cm) | $41.0 \pm 8.9$ | $41.2 \pm 14.5$ | $25.1 \pm 12.1$ | $32.3 \pm 16.6$ | $21.8 \pm 11.2$ | $27.2 \pm 10.0$ | $17.6 \pm 6.4$ | $28.0 \pm 7.5$ |
| $BA_{tot}$ (m$^2$/ha) | 77.7 | 36.8 | 29.6 | 28.0 | 31.2 | 53.3 | 48.3 | 61.3 |
| $V_{tot}$ (m$^3$/ha) | 745.3 | 369.0 | 268.6 | 269.1 | 270.6 | 479.8 | 378.6 | 558.2 |
| Alders total | | | | | | | | |
| Density (n/ha) | 589 | 279 | 597 | 326 | 716 | 836 | 1989 | 995 |
| $DBH_{mean}$ (cm) | $41.0 \pm 8.9$ | $40.2 \pm 13.7$ | $25.1 \pm 12.1$ | $29.9 \pm 13.7$ | $20.4 \pm 10.6$ | $27.9 \pm 8.9$ | $17.6 \pm 6.4$ | $28.0 \pm 7.5$ |
| $BA_{tot}$ (m$^2$/ha) | 77.7 | 35.3 | 29.6 | 23.0 | 23.4 | 51.0 | 48.3 | 61.3 |
| $V_{tot}$ (m$^3$/ha) | 745.3 | 338.1 | 268.6 | 214.3 | 196.5 | 455.2 | 378.6 | 558.2 |
| Alive alders | | | | | | | | |
| Density (n/ha) | 557 | 279 | 597 | 326 | 716 | 820 | 1870 | 971 |
| $DBH_{mean}$ (cm) | $41.5 \pm 8.6$ | $40.2 \pm 13.7$ | $25.1 \pm 12.1$ | $29.9 \pm 13.7$ | $20.4 \pm 10.6$ | $28.0 \pm 8.9$ | $17.9 \pm 6.5$ | $28.1 \pm 7.4$ |
| $BA_{tot}$ (m$^2$/ha) | 75.2 | 35.3 | 29.6 | 23.0 | 23.4 | 50.6 | 47.0 | 60.2 |
| $V_{tot}$ (m$^3$/ha) | 720.7 | 338.1 | 268.5 | 214.3 | 196.5 | 450.9 | 371.2 | 549.1 |
| Dead alders | | | | | | | | |
| Density (n/ha) | 32 | 0 | 0 | 0 | 0 | 16 | 119 | 24 |
| $DBH_{mean}$ (cm) | $31.5 \pm 8.9$ | 0 | 0 | 0 | 0 | $19.0 \pm 6.4$ | $11.6 \pm 2.7$ | $24.1 \pm 11.0$ |
| $BA_{tot}$ (m$^2$/ha) | 2.47 | 0 | 0 | 0 | 0 | 0.45 | 1.27 | 1.09 |
| $V_{tot}$ (m$^3$/ha) | 24.6 | 0 | 0 | 0 | 0 | 4.3 | 7.4 | 9.1 |
| Other trees | | | | | | | | |
| Density (n/ha) | 0 | 8 | 0 | 16 | 119 | 80 | 0 | 0 |
| $DBH_{mean}$ (cm) | 0 | $69.0 \pm 0.0$ | 0 | $63.5 \pm 5.4$ | $28.9 \pm 12.7$ | $19.2 \pm 14.1$ | 0 | 0 |
| $BA_{tot}$ (m$^2$/ha) | 0 | 1.6 | 0 | 5.0 | 7.9 | 2.3 | 0 | 0 |
| $V_{tot}$ (m$^3$/ha) | 0 | 30.9 | 0 | 54.8 | 74.2 | 19.6 | 0 | 0 |

### 3.3. Landscape Configuration

At a large scale, woodland was the prevalent cover type from the 200 m to 1000 m buffer for On2 and On3, while grassland prevailed for On5 and On6. At a smaller scale, in the 25 m buffer, the percentage of alder cover type was always over 90% for compact alder patches, while it ranged between 50% and 90% among the linear ones. Alder surface was less than 50% only for the On8 site. Moving from the 25 to the 50 m buffer, the alder area proportions did not vary significantly for On4 and On6 among the linear and for the compact On2. Among compact alder woodlots, On5 was associated with the greatest increase in grassland as the buffer radius increased. In the 50 m radius buffer, grassland prevailed for all the linear patches. Cover type percentage was quite constant for linear woodlots, while for compact ones, it varied significantly for On5 and On7. Alder percentage variation became irrelevant beyond the 200 m buffer. Ecotone forest/grassland was similar for the same buffer radius for all the sites, except for the 50 m buffer, where edge length was longer in linear than in compact woodlots (Table 5).

**Table 5.** Landscape analysis performed around light trapping points at different buffers. Total forest cover at 25 and 50 m of buffers corresponded to alder forest cover and was not duplicated in the table.

| Shape of Alder Woodlots | Linear | | | | Compact | | | |
|---|---|---|---|---|---|---|---|---|
| Woodlot | On1 | On4 | On6 | On8 | On2 | On3 | On5 | On7 |
| | | | Landscape composition | | | | | |
| | | | Grassland cover (%) | | | | | |
| buffer_25 | 14 | 46 | 46 | 59 | 0 | 0 | 3 | 0 |
| buffer_50 | 50 | 68 | 68 | 77 | 0 | 0 | 40 | 0 |
| buffer_200 | 62 | 62 | 92 | 58 | 9 | 1 | 82 | 29 |
| buffer_500 | 40 | 36 | 92 | 53 | 19 | 0 | 92 | 51 |
| buffer_1000 | 24 | 27 | 92 | 40 | 18 | 1 | 86 | 32 |
| | | | Alder forest cover (%) | | | | | |
| buffer_25 | 86 | 54 | 54 | 41 | 100 | 100 | 97 | 100 |
| buffer_50 | 50 | 31 | 32 | 23 | 100 | 84 | 60 | 74 |
| buffer_200 | 7 | 6 | 8 | 9 | 32 | 22 | 18 | 7 |
| buffer_500 | 6 | 4 | 6 | 3 | 7 | 7 | 7 | 3 |
| buffer_1000 | 2 | 1 | 3 | 2 | 2 | 3 | 5 | 1 |
| | | | Total forest cover (%) | | | | | |
| buffer_200 | 38 | 38 | 8 | 42 | 91 | 99 | 18 | 71 |
| buffer_500 | 60 | 64 | 8 | 47 | 80 | 100 | 8 | 49 |
| buffer_1000 | 76 | 73 | 8 | 60 | 82 | 99 | 14 | 68 |
| | | | Ecotone forests/grasslands (m) | | | | | |
| buffer_50 | 205 | 210 | 230 | 242 | 0 | 0 | 196 | 0 |
| buffer_200 | 1490 | 1584 | 739 | 1453 | 915 | 225 | 1647 | 1292 |
| buffer_500 | 5697 | 7731 | 4267 | 6097 | 3496 | 375 | 4236 | 7176 |
| buffer_1000 | 1365 | 17,436 | 13,220 | 22,135 | 10,833 | 1524 | 22,642 | 17,279 |

### 3.4. Moth–Forest Relationship

The moth–forest relationship was investigated, and we searched for correlations of moth data with the forest structure and landscape configuration of the sampled woodlot. Since the values of the alder plots on Axis 1 were very similar to each other, it was not useful to correlate these values with the values of Axis 1. We found very few significant correlations between forest structure and moth data. DBH and the linked estimated tree age were negatively correlated with species richness, whilst tree density was positively correlated with the compositional aspects of moth communities (CA Axis 2 values) and with moth abundance (Table 6).

**Table 6.** Linear correlations (Pearson) between moth community variables and structural attributes of forests. Only parameters with at least one significant correlation were included in the table. CA Axis 2: values of moth communities along Axis 2 of the Correspondence Analysis; $S$: number of species; $N$: number of individuals; $a$: Fisher's alpha diversity index.

| | CA Axis 2 | S | N | a |
|---|---|---|---|---|
| Forest total | | | | |
| Estimated age (y) | N.S. | −0.754 * | N.S. | N.S. |
| Density (n/ha) | −0.765 * | N.S. | N.S. | N.S. |
| $DBH_{mean}$ (cm) | N.S. | −0.784 * | N.S. | N.S. |
| Alders total | | | | |
| $DBH_{mean}$ (cm) | N.S. | −0.761 * | N.S. | N.S. |
| Alive alders | | | | |
| $DBH_{mean}$ (cm) | N.S. | −0.754 * | N.S. | N.S. |
| Other trees | | | | |
| Density (n/ha) | −0.821 * | N.S. | 0.825 * | N.S. |
| $DBH_{mean}$ (cm) | N.S. | N.S. | N.S. | −0.774 * |

$p < 0.05$ = *.

Community composition, synthesized by the values of CA Axis 2, was significantly related to selected landscape variables, mostly at the buffers of 50 and 200 m, but never to the largest one of 1000 m. Number of individuals was significantly correlated at the buffers of 50 and 200 m, and equitability was only correlated at the 500 m buffer. Grasslands and ecotones were positively correlated with moth compositional aspects (CA Axis 2 values) and equitability but negatively correlated with moth abundance. Consequently, the opposite occurred for forests (Table 7).

**Table 7.** Linear correlations (Pearson) between moth community variables and landscape configuration. Only parameters with at least one significant correlation were included in the table. CA Axis 2: values of moth communities along Axis 2 of the Correspondence Analysis; *N*: number of individuals; *J*: equitability.

|  | CA Axis 2 | *N* | *J* |
|---|---|---|---|
| Grasslands cover (%) | | | |
| buffer_25 | 0.719 * | N.S. | N.S. |
| buffer_50 | 0.898 ** | −0.760 * | N.S. |
| buffer_200 | 0.932 *** | N.S. | N.S. |
| buffer_500 | 0.786 * | N.S. | 0.732 * |
| Alder forest cover (%) | | | |
| buffer_25 | −0.719 * | N.S. | N.S. |
| buffer_50 | −0.893 ** | 0.847 ** | N.S. |
| buffer_200 | N.S. | 0.910 ** | N.S. |
| Forest cover (%) | | | |
| buffer_200 | −0.932 *** | N.S. | N.S. |
| buffer_500 | −0.784 * | N.S. | −0.732 * |
| Ecotone forests/grasslands (m) | | | |
| buffer_50 | 0.928 *** | −0.709 * | N.S. |

$p < 0.05$ = *; $p < 0.01$ = **; $p < 0.001$ = ***.

## 4. Discussion

Black alder forests inhabited well-characterized moth communities, and these results were very different from those found in a pine forest located at the same altitude, on the same geological substratum, and near the sampled alder woodlots. This occurred despite black alder forests usually being surrounded by pine forests. Ienco et al. [29] demonstrated that community composition is mainly driven by vegetation type, but our data also demonstrated the important roles of other drivers when an individual forest type is analyzed.

The study concerned eight sampled woodlots, which is considered a low number of sites, but it was very hard to find, in the field woodlots, pairs with the shape being the only difference. Furthermore, the dataset we used covered the whole flight period of moths encompassing the main beta-diversity driver for Lepidoptera, i.e., phenological changes [63], resulting in well-established datasets.

We found that moth communities of black alder forests of the Sila Massif were primarily shaped by landscape configuration, with forest structure being a weaker effect. Furthermore, our hypothesis that linear woodlots inhabit an impoverished version of communities inhabiting compact ones seemed to not be confirmed.

As expected for impoverished communities, a lower species richness resulted within linear moth communities, thus apparently supporting our main hypothesis. On the contrary, the NODF analysis failed to support it, as most of the linear communities were shaped by the turnover, with only one exception. The latter concerned a woodlot surrounded by a heavy, grazed grassland, where cows were present constantly across the summer. Grazing is known to be detrimental to moth diversity [64,65]. In our case, the impoverished moth community of the grassland was not able to support the turnover, increasing the contribution of nestedness. As a result, the turnover was the main beta-diversity process

involved in shaping linear woodlot communities, with grassland species entering the community in substitution of forest ones when meadows were well-preserved.

In multivariate analysis, moth communities were mostly grouped according to forest patch shapes, with one exception concerning the sole compact woodlot surrounded by grasslands (On5). We defined as compact the woodlots that were at least 50 meters wide, but our results demonstrated this measure to be too small for reducing the edge effect when a riparian woodlot is not surrounded by other forest types. In fact, the On5 community results were richer than expected in grassland individuals and poorer in forest ones. As an example, *Agrotis cinerea*, very abundant in mountain grasslands of south Italy [66], was particularly common within the On5 community. On the other hand, *Diarsia mendica*, living within different kinds of forests [67] and common within compact alder woodlots, was found with only one individual in this woodlot. It has already been observed that the increasing of edges, a proxy of forest fragmentation, leads to a decrease in forest species populations [12,68,69]. However, the edges do not all have the same ecological significance, as those between different forest types are obviously milder than those between forest and non-forest habitats, due to the similar abiotic conditions. As a result, the edges between forests are weaker barriers to moth movements, also allowing the persistence of a characteristic alder moth community in the case of small woodlots.

Only a few of the forest structure variables we studied were correlated with moth community attributes, i.e., estimated tree age, tree density, and diameter at breast height (DBH). Tree age and the related DBH were negatively correlated with diversity, apparently in contradiction with previous papers that found a higher diversity in older forests [70,71]. However, we found this discrepancy because trees growing along riverbanks and then composing linear woodlots were older than those growing within compact woodlots. This finding points out that a forest ecosystem can be defined as old not only when composed of old trees but also when all the portions of a given ecosystem are at a mature successional stage, as observed in old-growth forests [72].

Tree density seemed to be effective at determining moth community composition, as previously observed for other animals [73–75], due to changes in biotic and abiotic parameters. In addition, we found a positive correlation between tree density and moth abundance, as also observed by Fuentes-Montemayor et al. [13,75]. In denser forests, we would expect a reduced abundance of moths because the abundance and diversity of larval food plants is reduced due to low light [76–78]. On the contrary, we found an increased abundance of moths. This anomaly could be due to the lower predation pressure of Chiroptera registered in denser forests [79–81], but this should be better investigated.

Our data confirmed the findings of previous studies, in which a buffer of about 200 m was the best for describing the relationships of the Lepidoptera community composition with landscape metrics [82–84], with the relationships always being insignificant at 1000 m of buffer. Black alder forests usually occupy very small surfaces [85], with a decreasing relationship with moth community composition when the buffer under consideration is too large. However, considering the surfaces of all forest types, the correlation between forest cover and moth community composition is still very significant. The negative correlation we found between equitability and forest cover at 500 m of buffer could be due to the increase in the abundance of individual species linked to this habitat.

From a practical point of view, our study suggests that a width of 50 m may not be enough to give functionality to isolated riparian woodland patches, at least for moth communities. Our outcome may help not only forest habitat restoration planners aiming at increasing landscape connectivity [75] but also policy makers. For Italian law (D.Lgs 3 April 2018 n. 34, Testo Unico in materia di Foreste e Filiere Forestali [Consolidated Act on Forestry and Forest Chain]), a forest, to be such, must have a minimum width of 20 m, significantly narrower than needed, according to our results. Narrow and isolated forest patches can play, at their best, only the role of corridors and/or stepping stones for forest specialist species [58]. Maintenance of large riparian forests promotes the integrity of waterways [86], being beneficial for both biodiversity and water quality.

## 5. Conclusions

Riparian forests provide several ecological services, among which biodiversity conservation is of relevant importance, especially within national parks. However, in times of rapid habitat degradation, a more effective and informed riparian woodlot management is needed, and it should be supported by inter-disciplinary analyses. In this paper, we found a secondary role of forest structure in driving the composition and diversity of moth communities, with patch shape and landscape configuration being major roles. Differences between linear and compact woodlot pairs were mostly due to the turnover, not confirming the original hypothesis that linear woodlots inhabit an impoverished version of moth communities but only when grasslands are in a good conservation status. Our results can help improve forest management planning because of the importance arising from woodlot width and landscape configuration. Silvicultural intervention and forest restoration should avoid isolated spotted forest patches, since they may not deliver the expected ecological benefit, being unable to support a forest specialist community. Increasing the minimum sizes for a forest, to be considered as such, leads to reducing the edge effect, especially when other forests do not surround the riparian woodlot. This may assume an even greater significance in times of rapid landform transformations and climate changes that are expected to make riparian woodland more and more vulnerable.

**Supplementary Materials:** The following supporting information can be downloaded at: https://www.mdpi.com/article/10.3390/land12091670/s1, Table S1: Species stand matrix with number of individuals as abundance value.

**Author Contributions:** Conceptualization, S.S.; methodology, V.B., R.T. and S.S.; formal analysis, V.B., R.T., F.F. and M.L.; data curation, V.B., R.T., L.B. and S.S.; writing—original draft preparation, S.L.C., M.L. and S.S.; writing—review and editing, F.F., G.R., L.B., G.Z., S.L.C., M.L. and S.S.; visualization, M.L. and F.F.; supervision, S.S.; funding acquisition, S.S. All authors have read and agreed to the published version of the manuscript.

**Funding:** This research was funded by the Sila National Park, Project "Il Barcoding delle farfalle nel Parco Nazionale della Sila: aree umide", with the support of the National Biodiversity Future Centre (NBFC) to Research Centre for Forestry and Wood, Rende, funded by the Italian Ministry of University and Research, PNRR, Missione 4 Componente 2, "Dalla ricerca all'impresa", Investimento 1.4, Project CN00000033, and Agritech National Research Center European Union NextGenerationEU (PIANO NAZIONALE DI RIPRESA E RESILIENZA (PNRR)—MISSIONE 4 COMPONENTE 2, INVESTIMENTO 1.4—D.D 1032 17/06/2022, CN00000022).

**Data Availability Statement:** Data available from: Leonetti, F.L.; Greco, S.; Ienco, A.; and Scalercio, S. Lepidopterological fauna of *Alnus glutinosa* (L.) Gaertn., forests in the Sila Massif (southern Italy) (Insecta: Lepidoptera). *SHILAP Revista de lepidopterología* **2019**, *47*, 535–556.

**Acknowledgments:** We would like to thank Marco Infusino, Silvia Greco, and Carlo Di Marco for their help during field surveys and Giuseppe Luzzi (Sila National Park) for supporting this study by releasing the collecting permit.

**Conflicts of Interest:** The authors declare no conflict of interest.

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
