# Peer review of "Compact vs. Linear: Effects of Forest Structure, Patch Shape and Landscape Configuration on Black Alder Macromoth Communities"

_land, doi:10.3390/land12091670_

Round 1

Reviewer 1 Report

The paper by La Cava et. al. analyzed the response of moth communities to landscape and forest structure in Black Alder forest in Sila National Park, Italy. The study aims to discuss the environmental factors affecting the moth diversity pattern in a forest landscape. This is a important study for Insect community ecology. The results are also interesting! ! For the most part, the writing flowed well and interpretation was easy, but I did get bogged down a few times in the details of the results section. So I suggest a few things below. Results: 1. In table 3, what is the "cfr." mean in "Hoplodrina cfr. octogenaria"? 2. In section 3.4, why didn't analysis the relationship of moth and CA axis 1?

Author Response

Replay to Reviewer 1

Point 1

In table 3, what is the "cfr." mean in "Hoplodrina cfr. octogenaria"?

Thanks for your warning. We added the meaning of “cfr."

Point 2

In section 3.4, why didn't analysis the relationship of moth and CA axis 1?

Thanks for your comment. We provided a brief explaination for the reason why we didn’t analysis the relationship between moths and CA axis 1.

Reviewer 2 Report

The article is devoted to the influence of the shape of forests on biodiversity. The problem is described on the example of the fauna of moths. The study can be attributed to the field of landscape ecology. The authors mainly used data from Leonetti et al. In addition, the authors collected their data using the collection methodology of Leonetti et al. The data were analyzed using both statistical methods and GIS.

The collection of initial data in the course of field work does not cause any complaints. Questions arise about the use of GIS and the measurement of landscape configuration:

1) You used polygon area and border length. Why did you choose these characteristics? Are they chosen as the most simple and sufficient for solving the tasks? Have you considered using other landscape metrics, such as those available in the LecoS plugin for QGIS?

2) What satellite image did you use to map the forests? Is this a single image for which the satellite and date of shooting are known? Or is it a mosaic of satellite images that you uploaded online to QGIS? If this is a mosaic, then you need to specify what kind of mosaic it is and what plugin it was loaded with.

3) In subsection 2.4, you need to indicate which version of QGIS was used.

4) In line 164, you need to remove the typo. Instead of QGis there should be QGIS.

The results obtained and their interpretation do not cause any complaints. The conclusions seem reasonable. I believe that the article can be published after making the above small changes.

Author Response

Reply to Reviewer 2

Point 1

You used polygon area and border length. Why did you choose these characteristics? Are they chosen as the most simple and sufficient for solving the tasks? Have you considered using other landscape metrics, such as those available in the LecoS plugin for QGIS?

Thanks for your comments. We added some few lines to clarify our choice. Substantially areas and perimeters represented the basic information to compute cover type fractions (or proportions among cover types) and edge lenght, which were selected among the several landscape metrics as the simplest measures of 1) landscape composition (cover type-specific) and 2) spatial configuration. We think it is unnecessary to specify that for the first task the use of other metrics such as “Dominance” or “Diversity” would not have been representantive of the specific cover type while for the second one we were interested just in the edge lenght indipendently from its distribution. Finally, the computation of more complex metrics would have been beyond the scope of the work.

Point 2

What satellite image did you use to map the forests? Is this a single image for which the satellite and date of shooting are known? Or is it a mosaic of satellite images that you uploaded online to QGIS? If this is a mosaic, then you need to specify what kind of mosaic it is and what plugin it was loaded with.

Thanks for your suggestions. We added information about source and date of satellite images and about the plugin that we used to upload them.

Point 3

In subsection 2.4, you need to indicate which version of QGIS was used.

Thanks for your warning. We provided information about the version of QGIS

Point 4

In line 164, you need to remove the typo. Instead of QGis there should be QGIS.

Thanks. Done.